# Impact of Equine Ocular Surface Squamous Neoplasia on Interactions between Ocular Transcriptome and Microbiome

**DOI:** 10.3390/vetsci11040167

**Published:** 2024-04-07

**Authors:** Lyndah Chow, Edward Flaherty, Lynn Pezzanite, Maggie Williams, Steven Dow, Kathryn Wotman

**Affiliations:** 1Department of Clinical Sciences, College of Veterinary Medicine and Biomedical Sciences, Colorado State University, Fort Collins, CO 80523, USA; lyndah.chow@colostate.edu (L.C.); edward.flaherty@colostate.edu (E.F.); lynn.pezzanite@colostate.edu (L.P.); maggie.mary.williams@colostate.edu (M.W.); 2Department of Microbiology, Immunology, and Pathology, College of Veterinary Medicine and Biomedical Sciences, Colorado State University, Fort Collins, CO 80523, USA

**Keywords:** ocular surface squamous neoplasia, horse, microbiome, transcriptome

## Abstract

**Simple Summary:**

Ocular surface squamous neoplasia (OSSN) is a common corneal and conjunctival cancer in horses living in regions of high UV exposure. The response to various therapies remains highly variable (30 to 90% response rates described in previous publications), and recurrence remains relatively common. Improved understanding of ocular transcriptomic responses to OSSN in horses, along with microbiome changes and how the ocular transcriptome and microbiome interact, may provide new insights into the disease pathogenesis and new leads for treatment. In the present study, we used swabs from the ventral conjunctival fornix of the OSSN-affected eye and the normal opposite eye to interrogate both the transcriptome and microbiome from the same sample. We then used bioinformatic approaches to identify key conjunctival cell-microbiome interactions and how these were affected by the presence of OSSN.

**Abstract:**

Ocular surface squamous neoplasia (OSSN) represents the most common conjunctival tumor in horses and frequently results in vision loss and surgical removal of the affected globe. Multiple etiologic factors have been identified as contributing to OSSN progression, including solar radiation exposure, genetic mutations, and a lack of periocular pigmentation. Response to conventional treatments has been highly variable, though our recent work indicates that these tumors are highly responsive to local immunotherapy. In the present study, we extended our investigation of OSSN in horses to better understand how the ocular transcriptome responds to the presence of the tumor and how the ocular surface microbiome may also be altered by the presence of cancer. Therefore, we collected swabs from the ventral conjunctival fornix from 22 eyes in this study (11 with cytologically or histologically confirmed OSSN and 11 healthy eyes from the same horses) and performed RNA sequencing and 16S microbial sequencing using the same samples. Microbial 16s DNA sequencing and bulk RNA sequencing were both conducted using an Illumina-based platform. In eyes with OSSN, we observed significantly upregulated expression of genes and pathways associated with inflammation, particularly interferon. Microbial diversity was significantly reduced in conjunctival swabs from horses with OSSN. We also performed interactome analysis and found that three bacterial taxa (*Actinobacillus*, *Helcococcus* and *Parvimona*) had significant correlations with more than 100 upregulated genes in samples from animals with OSSN. These findings highlight the inflammatory nature of OSSN in horses and provide important new insights into how the host ocular surface interacts with certain microbial populations. These findings suggest new strategies for the management of OSSN in horses, which may entail immunotherapy in combination with ocular surface probiotics or prebiotics to help normalize ocular cell and microbe interactions.

## 1. Introduction

Ocular surface squamous neoplasia (OSSN) is the most common conjunctival tumor in horses, frequently resulting in vision loss and potentially surgical removal of the globe [1,2,3]. Risk factors for the development of OSSN include lack of periocular pigmentation, solar radiation exposure, heritable genetic mutations, acquired p53 mutations, physical periocular irritation, previous viral exposure, and potentially immunological and hormonal influences [4,5,6,7,8,9,10,11,12]. Although multiple treatment options have been described (e.g., surgical removal, intralesional chemotherapy, radiation therapy, cryotherapy, radiofrequency hyperthermia, immunotherapy, or combination), success rates vary widely (30 to 90%), and recurrence has been observed with all available treatment modalities [5,6,13,14,15,16,17,18,19,20,21,22,23,24,25,26,27]. Recently, however, the application of a novel liposomal immunotherapy has substantially improved OSSN treatment responses (including eyelid tumors) while minimizing local toxicity [27]. The ocular pathology of equine OSSN also presents several key similarities to that of OSSN in humans, including shared risk factors, lesion progression time, and similar gross and histologic appearances [1,9,10], further supporting the additional translational value of studies expanding our mechanistic understanding of OSSN disease processes in the equine model.

Pathophysiological changes in the tumor microenvironment, including the microbiome, have been shown to significantly impact tumor growth [28,29,30,31]. Microbiota exert significant effects on the host immune system, metabolism and disease progression [32,33]. The role of the microbiome in tumor initiation, prognosis, and response to treatment has been a recent topic of discussion, which has largely focused on the relationship between the gastrointestinal microbiota dysbiosis and cancer progression but has more recently expanded to explore the concept that the local tumor microbiota likely plays an integral role in disease progression [34,35]. Although the bacterial load on the ocular surface is low, in humans, these bacteria are thought to play a role in maintaining corneal homeostasis and regulating the immune system by stimulating host interleukin production and activation [36]. The gut microbiome has been recognized to indirectly modulate cancer susceptibility or progression, while shifts in intra- or peri-tumoral microbiota diversity have further been demonstrated in multiple other cancer types in humans, including colorectal, lung, breast, prostate and cervical [37,38,39,40,41,42,43,44,45,46,47]. Using in situ spatial-profiling technologies and single-cell RNA-sequencing of human oral squamous cell carcinoma (SCC) and colorectal cancers, spatial, cellular, and molecular host-microbe interactions were recently demonstrated [46]. Although previous investigations of alterations in cellular and molecular biology have advanced earlier cancer diagnosis and treatment in horses, microbial contributions to tumor growth and host differential gene expression and the potential therapeutic implications of such characterization remain incompletely unexplored [45].

A unique aspect of ocular anatomy is the capability of studying animals with unilateral ophthalmic disease, allowing the non-affected eye to act as a “control” due to its exposure to the same environmental conditions as the affected eye. Therefore, the objective of this study was to characterize the local microbiota of the conjunctiva in equine eyes affected by OSSN compared to healthy eyes and to associate the microbiota with tumor RNA expression profiles. We documented the most prevalent species observed in ocular tissues and assembled preliminary evidence supporting microbial compositional shifts in ocular neoplasia. We used microbial read evidence and host transcriptional expression from the same tissues to perform association analyses, representing the first study to examine both microbial presence and gene expression from the same prepared sampling sites in equine neoplasia with translational value to humans suffering from similar OSSN disease processes.

## 2. Materials and Methods

### 2.1. Horses

The clinical studies described here were approved by the Colorado State University Clinical Review Board (protocol #3556). Horses enrolled were presented to Colorado State University’s Johnson Family Equine Hospital Equine Ophthalmology Service between March 2022 and June 2023. Horses were screened prospectively by complete ophthalmic examination, including biomicroscopy, direct ophthalmoscopy, and tonometry, as well as patient history, physical examination, and bloodwork to ensure no prior history of systemic disease, and enrolled with informed owner consent. Horses were included if no concurrent systemic disease was detected and following histopathologic or cytologic confirmation of OSSN with the tumor localized to the limbus. Horses were excluded if lesions of the primary eyelid or third eyelids were noted. A total of 11 horses (22 eyes) were included in the study, including 1 mare, 9 geldings, and 1 stallion. Horse breeds included four Paint Horses, three Quarter Horses, one Thoroughbred, one Tennessee Walking Horse, one Cob horse, and one Mustang. Of the 11 OSSN-affected eyes sampled, OSSN was present in the right eye of 7 horses, and the left eye of 4 horses. The perilimbal location of the OSSN in affected eyes were lateral (8/11), medial (2/11), and ventral (1/11). Affected horses presented at an average age of 16.6 years old (range 5 to 24 years). Lesions were noted either by the owner or a referring veterinarian between 3 weeks and 10 months prior to the initial presentation.

### 2.2. Sample Collection and Processing

Ocular surface swabs were collected using 6-inch PurFlock Ultra and sterile flocked collection devices (Puritan Medical Products, Guilford, ME, USA). The horse’s inferior eyelid was everted, the dry swab was inserted to the level of the ventral conjunctival fornix and swept along the length of the eyelid four times. Three swabs were placed in a 15 mL conical tube (Corning Inc., Corning, NY, USA) containing RNAlater (ThermoFisher Scientific, Waltham, MA, USA) for RNA sequencing, while one swab was placed in a micro conical tube without any storage media for microbial 16S sequencing. Samples for RNA sequencing were stored at 4C prior to RNA and DNA extraction. To extract RNA from nasal cells, swabs were vortexed for 1 min at high speed to dislodge cells. PBS was then added to the RNA later at a ratio of 1:5. Cells were pelleted, then processed for RNA extraction using the Qiagen RNeasy micro kit (Qiagen, Hilden, Germany) following manufactures instructions for DNA extraction as described below.

### 2.3. RNA Sequencing

RNA concentrations were verified on a Nanodrop 1000 Spectrophotometer (ThermoFisher Scientific) and then sent to Novogene Corp. Inc. (Sacramento, CA, USA) for RNA sequencing. RNA quality was determined using an Agilent 2100 Bioanalyzer system to generate RIN numbers (RNA integrity number), which ranged from 6.9 to 10 for all RNA samples submitted. At Novogene Corp, mRNA was purified using poly-T oligo-attached magnetic beads. After fragmentation, the first strand of cDNA was synthesized using random hexamer primers, followed by the second strand of cDNA synthesis. The library was completed following end repair, A-tailing, adapter ligation, size selection, amplification, and purification. Quantified libraries were pooled and sequenced on an Illumina NovaSeq 6000 (Illumina, San Diego, CA, USA). 150 bp paired-end reads were generated, and files were delivered as de-multiplexed fasq files.

Sequence data were analyzed on Partek Flow software, version 10.0 (Partek Inc., Chesterfield, MO, USA). Raw data were filtered by removing reads containing adapters and reads containing N > 10% and for Phred scores > 30. Filtered reads were aligned with STAR 2.7.3a using the CanFam3.1 genome assembly. Aligned reads were annotated and counted using HT-seq [48] with Ensembl 107, and differentially expressed genes were identified using DEseq2 [49] (Differential gene expression analysis based on negative binomial distribution). Biological interpretations included gene ontology and gene set enrichment analysis (GSEA), (https://www.gsea-msigdb.org/gsea/index.jsp, accessed on 1 December 2023). Gene sets Hallmarks v2022.1, biocarta v2022.1, KEGG v2022.1, Gene Ontology go.bp v2022.1, and ImmuneSigDB v2022.1 were used for comparisons. Significant pathways were filtered using false discovery rate (FDR) q-value of ≤0.25 and NOM *p*-value ≤ 0.05.

### 2.4. Microbial 16S Sequencing

Additional swabs were collected for microbial analysis. Swabs were cut and immersed in extraction tubes following Qiagen DNeasy PowerSoil Pro Kit instructions (Qiagen, Hilden, Germany). Microbial DNA was frozen at −80 °C and sent to ACME (Anschutz Center for Microbiome Excellence) at the University of Colorado Anschutz Medical Campus, Aurora, CO. for microbial sequencing. The library was prepared according to the Earth Microbiome project protocol (https://earthmicrobiome.org/protocols-and-standards/16s/, 13 February 2024), with 35 PCR cycles using 515F and 806R primers. Due to the low overall bacterial abundance in the swab samples, additional PCR cycles were needed to reach the required number of reads for each sample. Samples were run on MiniSeq cartridges on Illumina Miseq sequencing instruments. Microbial sequence analyses were performed with QIIME2 [50]. Microbial community similarity was displayed with principal coordinate analysis (PCoA) plots. Alpha diversity was determined using Shannon, Faith, and pielou indices. Beta diversity using weighted and unweighted UniFrac, as well as Bray Curtis. Alpha diversity indices were compared using a paired T-test, and beta diversity metrics were compared with PERMANOVA. An analysis of the composition of microbiomes (ANCOM) was employed to determine the sequence variants that differed significantly between treatment groups [51]. In addition, LEfSE (Linear discriminant analysis Effect Size) was also used to calculate the taxa that best discriminated between rhinitis and the healthy group (https://huttenhower.sph.harvard.edu/lefse/, 13 February 2024) [52]. Microbial features were filtered for a minimum frequency of 22 (0.01% of the highest) and features not present in >2 samples were also removed, resulting in a total of 365 total features.

### 2.5. Interactome Analysis of Transcriptome and Microbiome Data

The microbial DNA of n = 6 out of 11 horses with matched RNA sequencing data were used in this analysis. The remaining horses had no microbial DNA extractions available. For this analysis, a total of 268 DEGs (differentially expressed genes) were extrapolated from the RNA sequencing data using a Log2 fold change of ≤−1 or ≥1. The median ratio normalized reads from individual samples were then correlated to the percent relative abundance of 212 unique bacterial taxon found in at least 3 samples using rcorr [52]. *p* values for significance and r values for correlation were generated for each gene to taxon pair. Protein-coding genes with correlation *p*-values ≤ 0.05 were then entered into the string protein database (https://string-db.org) for categorizing the protein sets [53]. 

## 3. Results

### 3.1. Transcriptomic Differences in OSSN Effected Surface Ocular Cells

RNA sequencing results from n = 11 ocular swabs with matched OSSN and normal phenotype show a high level of heterogeneity. Dimensional reduction shows no obvious clustering of OSSN samples; instead, 4 out of 11 of the OSSN-affected eyes are separated from the others (Figure 1A). Differential gene expression analysis shows 174 significantly upregulated protein-coding genes (Figure 1B). Highly upregulated genes in OSSN included GPCRs (G protein-coupled receptors), cancer-associated PTX and OSM, as well as the inflammatory cytokines IL1B and CXCL8 9 (Figure 1C). 

Gene set enrichment analysis (GSEA) comparing OSSN to normal eyes shows significant upregulation of multiple immune and inflammatory response pathways (Figure 2A,B), as well as the upregulation of reactome pathways related to interleukin signaling, platelets, neutrophils, and extracellular matrices (Figure 2D). Downregulated pathways in OSSN are less numerous and include “cilium assembly” as well as some RNA processing pathways (Figure 2C). 

### 3.2. Ocular Microbiome of OSSN-Affected Horses

The ocular surface microbiome was compared between n = 6 OSSN-affected eyes and matched normal eyes using 16S microbial DNA seq. After filtering (see methods), a total of 365 features were present in n = 12 samples. The most abundant phylum on average was *Proteobacteria*, followed by *Actinobacteria*, then *Firmicutes* (Figure 3A), while the most abundant taxa on a class level is *Gamaproteobacteria*, followed by *Actinobacteria* (class), then *Bacilli* and *Clostridia* (Figure 3B). OSSN-affected eyes showed an overall decrease in alpha diversity (Figure 3C); however, the *p*-value did not reach a significant number. PCoA (principal coordinate analysis) using weighed unifrac distance measurement showed clustering of five out of six OSSN samples, with a PERMANOVA (permutational multivariate analysis of variance) *p*-value of 0.053 (Figure 3D). There were no significant differences in composition at any taxon level between OSSN vs. normal eyes using ANCOM or LefSE 

### 3.3. Interactome of OSSN

In order to provide the most comprehensive and biologically relevant interactome composition, species and genus-level taxa were selected for correlation with DEGs from sample-matched RNA sequencing data. A total of 11 species were found to have significant correlations with at least one gene (Figure 4A). 

However, there were no major associations or common pathways found between these 21 genes. The genes KRT77 (Keratin 77) and GMNN (Geminin) had the most correlations (all positive) with 8 out of the 11 species. On a genus level, a total of 18 genus-level annotations were correlated to the 268 DEGs as described in methods. There were three genera with significant correlations to more than 100 genes, including *Actinobacillus*, *Helcococcus* and *Parvimonas* (Figure 4B). There were 242 genes with a significant correlation to *Actinobacillus*, mapping to STRING pathways; the highest-strength pathways corresponded to immune-related pathways, such as MHCI, IL8, IL1, IL18 and TLR pathways (Figure 4C). The highest-strength pathways for the 107 genes correlated to *Helcococcus* are similar to those of *Actinobacillus*, with additional macrophage proliferation and some Th1 responses (Figure 4D). There were 233 genes with a significant positive correlation to *Parvimonas*, with the highest-strength immune pathways mostly identical to *Actinobacillus* (Figure 4E). Both *Parvimonas* and *Helcococcus* had positive correlations with r values > 0.6, whereas *Helcococcus* was negatively correlated with 4 out of the 107 genes. 

## 4. Discussion

The primary objective of this study was to investigate how the transcriptome and the microbiome are altered in horses with OSSN and how these two unique areas of the ocular immune system interact with each other. This is the first study to our knowledge to explore the ocular transcriptome and microbiome using matched ocular swab samples, and the first to study their interactions in an equine ocular neoplasia model.

Based on the transcriptome analysis, we identified 174 genes that were significantly upregulated in the OSSN-affected eyes compared to the controls. To compare the ocular transcriptome of horses with OSSN to that of human OSSN, a query of several of these upregulated genes was submitted to the “human eye atlas” [54], in which seven OSSN-affected samples were included in the analysis. Many of the upregulated genes (e.g., OSM, IL-1ß, CXCL8, ORL1, PTGS2, TREM1) are upregulated in human hyalocytes, which are tissue-resident innate immune cell populations present in the vitreous cavity and thought to be derived from the monocytic lineage [55,56]. However, the pathophysiology of OSSN is traditionally recognized to originate in the limbal epithelial cells [57,58], and indeed Boneva et al. demonstrated an upregulation of many keratin and epidermal development genes upregulated in SCC tissues, which was not seen in our equine study. We did, however, observe a strong immune signature, as stated in the pathway analysis, for example, with upregulation of complement proteins, inflammatory cytokines and inflammatory response pathways (Figure 2). These conjunctival and corneal cellular responses in horses with OSSN likely reflect inflammation associated with immune responses to the malignant cells. For example, previous histological studies of equine OSSN have demonstrated an inflammatory cell infiltrate in most of the tumor biopsies that were examined, indicative of an ongoing anti-tumor immune response [59,60]. In addition to the plethora of immune pathways, cilium assembly was downregulated in OSSN eyes, which is in line with previous reports of dysregulation in ocular diseases [61]. 

We also assessed the ocular microbiome in this study to help understand not only how the microbiome may be altered in OSSN but also to interrogate using Next Gen Sequencing techniques the ocular microbiome in normal horses, which has been previously described in other geographic regions including Alabama, Texas, Chile, and the United Kingdom [62,63,64,65]. Across these studies, the most common phylum identified oscillates between *Proteobacteria*, *Actinobacteria*, *Firmicutes* and *Bacteroidetes*, with major differences attributed to geographic location. Julian et al. reported an increase in the class *Bacilli* (phylum *Firmicutes*) in the eyes of horses with ulcerative keratitis [66]. The human ocular microbiome is arguably less “complex” compared to horses living in an outdoor environment, composed primarily of *Proteobacteria* and *Actinobacteria* [67]. As in these previously reported human studies, the genus *Corynebacterium* (phylum *Actinobacteria*) was also present at high levels here in equine patients. With respect to OSSN, there is a single report in cattle that compares the conjunctival microbiome in eyes with squamous cell carcinoma to those of healthy eyes [68], which did not identify significant differences in bacterial populations.

In the present study, the OSSN microbiome alpha and beta diversity were significantly different in OSSN eyes, although there were no significant differences in microbial abundance between groups. Related to our findings, we noted from previous reports that the genus *Actinobacillus* has long been associated with various inflammatory and infectious diseases such as granuloma, sepsis, and arthritis in livestock [68]. In the current study, we found *Actinobacillus* abundance to be correlated with upregulation of many immune genes in the major histocompatibility complex (MHC), interleukin, and microglia categories (Figure 4), as was the case of the genus *Helcococcus*, which also has a strong “immune” correlation. Although levels of neither *Actinobacillus* or *Helcococcus* genus were significantly different between OSSN and normal eyes, *Actinobacillus* was highly enriched in four out of the six OSSN eyes, making it a more likely candidate for a disease-mediated process directly related to the presence of tumors. The genus *Parvimonas* was also correlated to the same categories of immune genes upregulated in OSSN eyes. *Parvimonas,* however, has previously been correlated to colorectal cancer [69]. In particular, *Parvimonas micra* is thought to epigenetically alter the methylation profile of tumor suppressor genes, thus contributing to metastasis [70]. Interestingly, microbial sequencing performed here also detected several species-level ASVs present. Since 16S sequencing uses only up to 250 base pairs, the species-level annotations are typically considered unreliable [71]. However, we were able to identify 13 species with >0.1% average abundance across all samples. Although the species-correlated genes did not all fall into any pathway classification or protein class, the correlation of eight of these species with the keratin gene KRT77 is intriguing, since KRT77 is considered a biomarker for squamous cell carcinoma [72] and involved in the epithelial differentiation from the limbal stem cells [73]. 

Although this study provides a novel glimpse into the OSSN immunome, there are several limitations that warrant additional discussion. As reports in the human literature indicate that the location of sampling (lid vs. conjunctiva) and extraction methods impact the microbial population sequences [74,75], the utmost care was taken to obtain samples from inferior conjunctival fornix; however, it is possible that some of the differences could be attributed to the swabbing technique. In canines with corneal ulcers, sampling from the conjunctival fornix yielded similar results in microorganism identification as sampling from the ulcer itself, suggesting the good utility of conjunctival fornix sampling as used in this study [76]. Similarly, the collection of host cells for transcriptome sequencing could also be impacted by the location of the swab and the cell types collected. As the sample size here was limited and as horses were client-owned with naturally occurring disease processes, horses presented at varying degrees of tumor severity, which may have also resulted in potential differences in the degree, stage or inflammation.

Our findings, which are the first to examine the ocular immune response in equine eyes with OSSN versus normal eyes, provide key new insights into the disease pathogenesis and present the possibility for a role of immunome in establishing or perpetuating the disease. Additional, larger studies will be required to establish a potential causative role more firmly for the microbiome in the host response to OSSN.

## 5. Conclusions

In this study, we demonstrated alterations in differential gene expression and the local microbiome in equine OSSN, with potential implications for disease susceptibility and progression. Key findings were that OSSN-affected eyes demonstrated a high level of heterogeneity in differential gene expression, with upregulation of genes associated with immune/inflammatory pathways associated with interleukin signaling, neutrophils, and the extracellular matrix. Evaluation of the local microbiome indicated a decrease in alpha diversity as well as differences in beta diversity in OSSN-affected eyes. Interactome analysis revealed significant interactions between the ocular cellular responses to OSSN and certain microbial populations, including *Actinobacillus*, *Helcococus*, and *Parvimonas*, suggesting important interactions between the host and the microbiome at the conjunctival epithelial surface.

## Figures and Tables

**Figure 1 vetsci-11-00167-f001:**
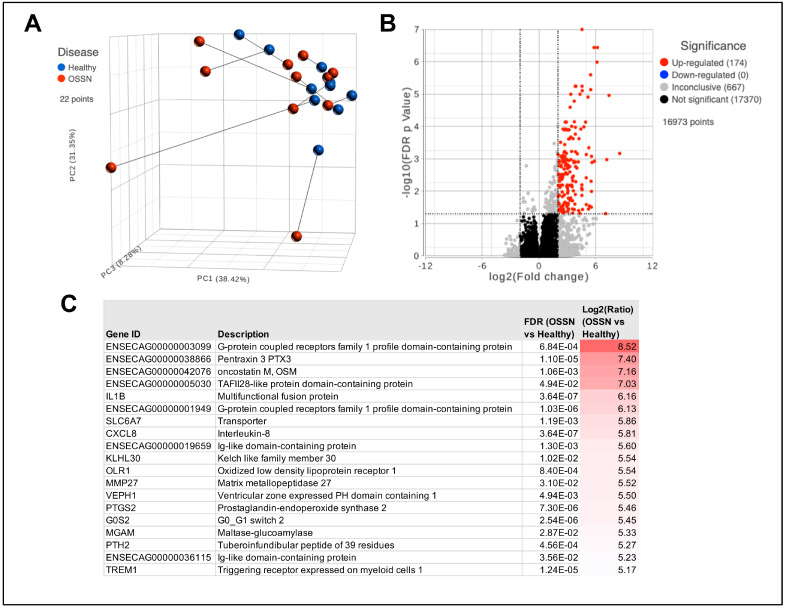
Transcriptomic differences in OSSN vs. normal ocular surface cells. (**A**) Principal component analysis (PCA) plot of n = 11 horses, with OSSN-affected eyes shown in red and normal matched horse eye shown in blue. (**B**) Volcano plot of Deseq differential gene expression analyses results. X-axis shows Log2 Fold change comparing OSSN vs. Normal and y-axis shows −Log10 false discovery rate adjusted *p*-value. Significance was defined as FDR ≤ 0.05 and fold change ≥ 2Log2 or ≤−2Log2. (**C**) List of top 20 significantly upregulated genes in OSSN-affected eyes with gene description, adjusted *p*-value, and fold change.

**Figure 2 vetsci-11-00167-f002:**
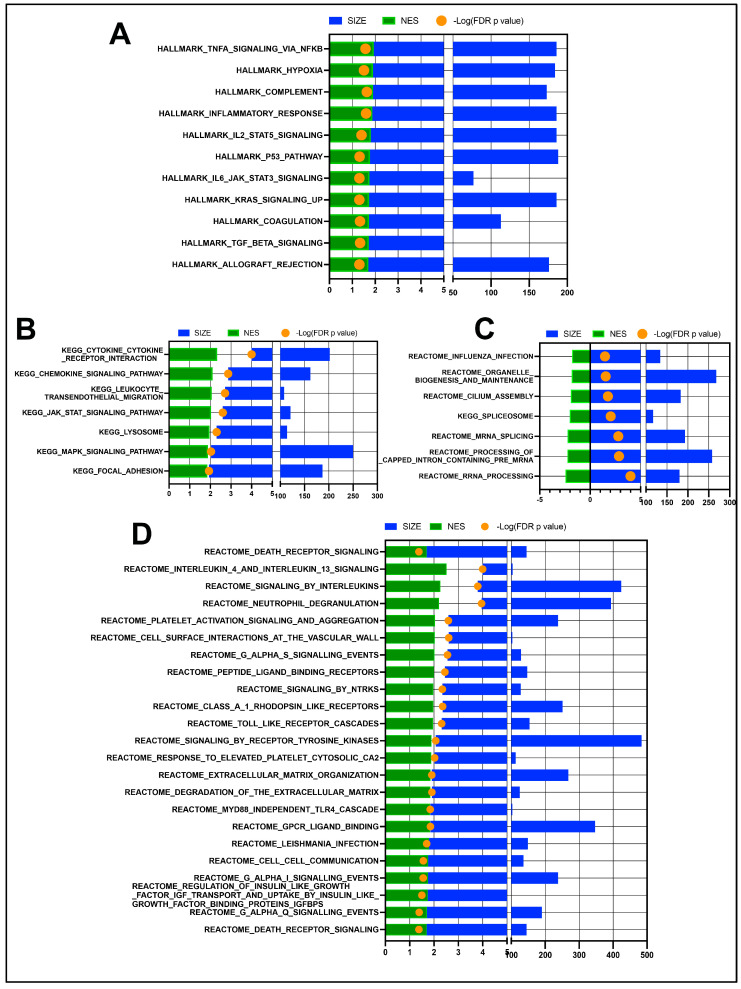
GSEA pathway analysis results of n = 11 OSSN-affected ocular swab cells vs. normal. (**A**) Stacked bar graph of Hallmark gene sets with FDR adjusted *p*-value ≤ 0.05. Normalized enrichment scores shown in green (positive for upregulated in OSSN), total genes mapped to pathways shown in blue and –log10(FDR *p*-value) in orange dot (>1.3 values significant). (**B**) Significant KEGG pathways upregulated in OSSN eyes. (**C**) significantly downregulated KEGG and Reactome pathways in OSSN. (**D**) additional significantly upregulated Reactome pathways in OSSN.

**Figure 3 vetsci-11-00167-f003:**
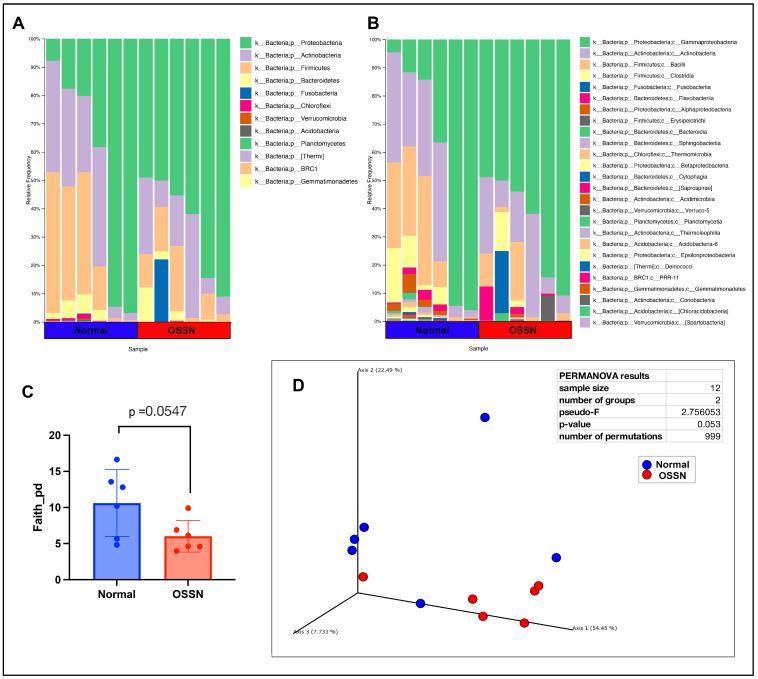
Ocular microbiome composition of OSSN-affected eyes compared to matched normal controls. (**A**) Taxonomy results of n = 6 horses, showing relative abundance at phylum level composition. (**B**) relative microbial abundance at class level showing a total of 26 classes of bacteria found in 12 samples. (**C**) Alpha diversity of OSSN vs. normal showing Faith metric with perceive distance (*p*) on y axis. Significance computed using paired parametric T test. (**D**) Beta diversity computed using weighed unifrac distance measurements. Normal samples in blue, OSSN in red. Statistical significance computed using QIIME2 permanova.

**Figure 4 vetsci-11-00167-f004:**
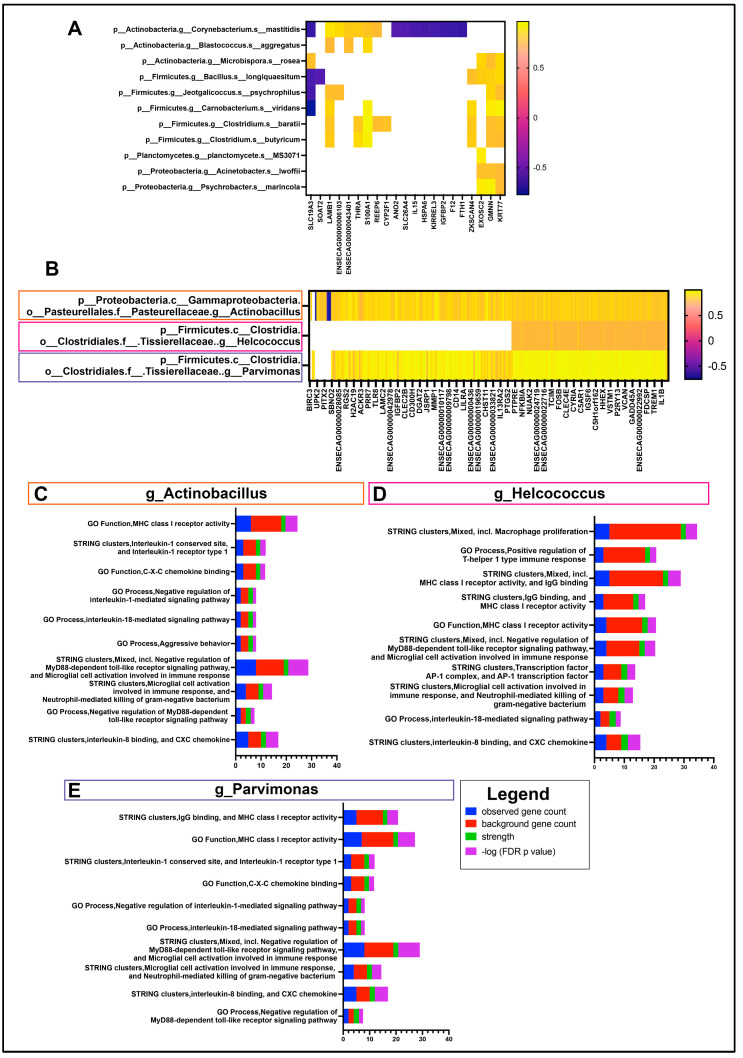
Ocular Interactome of equine OSSN. Differentially expressed genes (OSSN vs. normal) with fold change ≤ −1.5log2 or ≥1.5Log2 and FDR ≤ 0.05 were correlated with relative abundance of genus and species level microbial taxon. (**A**) shows significant correlations with 5 species and DEGs. r values shown in color scale. (**B**) Three genus-level taxon with >100 significant correlations with differentially expressed genes. Correlation r values shown in color scale, white spaces nonsignificant. (**C**) stacked bar graph of String protein annotation for DEGs significantly correlated with genus *Actinobacillus.* Pathways chosen with highest strength and *p* value ≤ 0.05. graph legend shown in bottom right. Blue shows number of DEGs in pathway, red shows total genes present in each pathway set. Green strength of association and purple for –Log10 FDR adjusted *p*-value. (**D**) top 10 strength String protein annotation pathways for DEGs correlated with genus *Helcococcus.* (**E**) top 10 strength String protein annotation pathways for DEGs correlated with genus *Parvimonas*.

## Data Availability

The datasets presented in this study will be made available in an online repository: GEO public genomic data repository, accession number pending.

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
