# Peer review of "Impact of Equine Ocular Surface Squamous Neoplasia on Interactions between Ocular Transcriptome and Microbiome"

_vetsci, 2024, doi:10.3390/vetsci11040167_

Round 1

Reviewer 1 Report

Comments and Suggestions for Authors

Very good article

Author Response

Thank you to the reviewer for their time reviewing the manuscript.

Reviewer 2 Report

Comments and Suggestions for Authors

The study entitled ‘Impact of equine ocular surface squamous neoplasia on 2 interactions between corneal transcriptome and microbiome’ sheds light on the complex interplay between equine OSSN, corneal transcriptome alterations, and microbiome changes, providing valuable insights into disease pathogenesis and potential therapeutic targets. The observed upregulation of genes associated with immune and inflammatory responses in OSSN-affected eyes suggests a role for these pathways in the disease process. The alterations in the local microbiome composition further emphasize the potential influence of microbial populations on disease susceptibility and progression. The significant interactions identified between corneal cellular responses and specific microbial species underscore the intricate relationship between the host and microbiome at the corneal surface.

I congratulate the authors for this valuable and pioneer research work. The manuscript is very well written and the contents clearly structured.

Some specific comments are listed below:

Title and Abstract:

The title effectively conveys the focus of the study on the impact of equine ocular surface squamous neoplasia on interactions between corneal transcriptome and microbiome. It accurately reflects the content of the research.

The abstract provides a clear and concise summary of the study, outlining the objectives, methods, key findings, and implications for future research. It effectively captures the essence of the research.

- Introduction:

The introduction sets the stage by providing a comprehensive background on equine ocular surface squamous neoplasia and the significance of understanding the interactions between the corneal transcriptome and microbiome in this context.

The research questions are clearly articulated, focusing on how the corneal transcriptome responds to the presence of the tumor and how the corneal surface microbiome may be altered by the neoplasia.

- Methodology:

The methodology section is well-detailed.

The use of RNA sequencing and microbial sequencing for corneal swabs is appropriately explained, including the platforms used for analysis.

- Results:

The results section effectively addresses the research questions, highlighting the significant upregulation of genes associated with inflammation in eyes with OSSN.

The findings on the reduction in microbial diversity in corneal swabs from horses with OSSN are well-supported by the data.

The interactome analysis revealing correlations between specific microbial taxa and upregulated genes in OSSN samples provides valuable insights into the host-microbiome interactions.

- Discussion:

The discussion interprets the results in the context of the existing literature, emphasizing the inflammatory nature of OSSN in horses and the potential implications for management strategies.

The authors acknowledge the limitations of the study, such as sample size and variations in tumor severity, and suggest avenues for future research to validate the findings.

The conclusions drawn align with the results presented, emphasizing the importance of understanding the host-microbiome interactions in equine OSSN for potential therapeutic interventions.

- References: the references cited are relevant and support the study's background and findings. However, it may be beneficial to include additional references to strengthen the points discussed in the penultimate paragraph of the discussion section.

- Overall evaluation: the study makes a significant contribution to the field of veterinary ophthalmology by elucidating the interactions between equine OSSN, corneal transcriptome alterations, and microbiome changes.

I recommend the publication of this manuscript after including additional references in the discussion section.

Author Response

Reviewer 2 –

The study entitled ‘Impact of equine ocular surface squamous neoplasia on 2 interactions between corneal transcriptome and microbiome’ sheds light on the complex interplay between equine OSSN, corneal transcriptome alterations, and microbiome changes, providing valuable insights into disease pathogenesis and potential therapeutic targets. The observed upregulation of genes associated with immune and inflammatory responses in OSSN-affected eyes suggests a role for these pathways in the disease process. The alterations in the local microbiome composition further emphasize the potential influence of microbial populations on disease susceptibility and progression. The significant interactions identified between corneal cellular responses and specific microbial species underscore the intricate relationship between the host and microbiome at the corneal surface. I congratulate the authors for this valuable and pioneer research work. The manuscript is very well written and the contents clearly structured. Some specific comments are listed below:

  • Thank you to the reviewer for their time and efforts towards improving the manuscript. The authors have responded to each comment below, with addition of references in the discussion section and other minor edits throughout the manuscript.

Title and Abstract: The title effectively conveys the focus of the study on the impact of equine ocular surface squamous neoplasia on interactions between corneal transcriptome and microbiome. It accurately reflects the content of the research.The abstract provides a clear and concise summary of the study, outlining the objectives, methods, key findings, and implications for future research. It effectively captures the essence of the research.

- Introduction: The introduction sets the stage by providing a comprehensive background on equine ocular surface squamous neoplasia and the significance of understanding the interactions between the corneal transcriptome and microbiome in this context. The research questions are clearly articulated, focusing on how the corneal transcriptome responds to the presence of the tumor and how the corneal surface microbiome may be altered by the neoplasia.

- Methodology: The methodology section is well-detailed. The use of RNA sequencing and microbial sequencing for corneal swabs is appropriately explained, including the platforms used for analysis.

- Results: The results section effectively addresses the research questions, highlighting the significant upregulation of genes associated with inflammation in eyes with OSSN. The findings on the reduction in microbial diversity in corneal swabs from horses with OSSN are well-supported by the data. The interactome analysis revealing correlations between specific microbial taxa and upregulated genes in OSSN samples provides valuable insights into the host-microbiome interactions.

  • Thank you for this feedback on the title, abstract, introduction, methodology and results sections.

- Discussion: The discussion interprets the results in the context of the existing literature, emphasizing the inflammatory nature of OSSN in horses and the potential implications for management strategies. The authors acknowledge the limitations of the study, such as sample size and variations in tumor severity, and suggest avenues for future research to validate the findings.The conclusions drawn align with the results presented, emphasizing the importance of understanding the host-microbiome interactions in equine OSSN for potential therapeutic interventions.

  • Thank you for this summary and feedback on presentation of the findings.

- References: the references cited are relevant and support the study's background and findings. However, it may be beneficial to include additional references to strengthen the points discussed in the penultimate paragraph of the discussion section.

  • Thank you for this suggestion – additional references have been added to strengthen the discussion section.

- Overall evaluation: the study makes a significant contribution to the field of veterinary ophthalmology by elucidating the interactions between equine OSSN, corneal transcriptome alterations, and microbiome changes. I recommend the publication of this manuscript after including additional references in the discussion section.

  • Thank you for this suggestion – additional references have been added to the discussion section.

Reviewer 3 Report

Comments and Suggestions for Authors THANK YOU FOR CHOOSING ME TO REVIEW THIS MANUSCRIPT. THE RESEARCH IS INTERESTING AND HAS A STRONG SCIENTIFIC IMPACT

Author Response

Thank you to the reviewer for their time and efforts reviewing the manuscript.